# Comprehensive Care and Education Can Improve Nutritional and Growth Outcomes in Children with Persistent Cow’s Milk, Egg, or Peanut Allergies: A Five-Year Follow-Up Study of Nutritional Status

**DOI:** 10.3390/nu16010048

**Published:** 2023-12-22

**Authors:** Tomaž Poredoš, Tina Vesel Tajnšek, Anja Koren Jeverica, Mojca Zajc Avramovič, Gašper Markelj, Nina Emeršič, Tadej Avčin

**Affiliations:** 1Department of Hospital Nutrition and Dietetics, University Children’s Hospital, University Medical Center Ljubljana, Bohoričeva 20, 1000 Ljubljana, Slovenia; 2Department of Allergology, Rheumatology and Clinical Immunology, University Children’s Hospital, University Medical Center Ljubljana, 1000 Ljubljana, Sloveniaanja.korenjeverica@kclj.si (A.K.J.); mojca.zajc.avramovic@kclj.si (M.Z.A.); gasper.markelj@kclj.si (G.M.); nina.emersic@kclj.si (N.E.); tadej.avcin@kclj.si (T.A.); 3Department of Paediatrics, Faculty of Medicine, University of Ljubljana, 1000 Ljubljana, Slovenia

**Keywords:** food allergy, children, adequate nutrition, catch-up growth

## Abstract

Background: Data suggest that food allergies greatly impact a child’s health and growth due to inadequate nutrient intake. Our study aimed to establish the long-term outcome of children with food allergies compared to a control group. Methods: This study was a retrospective cohort study with longitudinal follow-up with a mean period of 4.85 years from the diagnosis to the last study visit. The patients’ nutritional intake was assessed using a three-day food diary and analysed by a dietitian. Patients (61 boys and 33 girls, mean age 6.9 years) had a single food allergy including 21 patients with cow’s milk, 34 with egg, and 39 with peanut allergies. The control group included 36 children (19 boys and 17 girls, mean age 8.03 years). Blood analysis was performed on all participants. Results: Data from our study showed that patients with cow’s milk, egg or peanut allergies had normal growth and achieved catch-up growth from the diagnosis until the last study visit. In the cow’s milk allergy group, the allergy was shown to affect calcium intake (*p* < 0.05), while egg and peanut allergies did not impact the dietary intake of nutrients. None of the investigated food allergies affected blood results (*p* < 0.05). Conclusions: In the present study, we showed that single food allergies do not compromise growth in children if they are provided with appropriate support and that the affected children reach catch-up growth from the diagnosis.

## 1. Introduction

Food allergies have been shown to affect the lives of children adversely [1,2,3,4,5,6]. The basic management of food allergies is still based on food avoidance [7,8]. Therefore, the role of a dietitian in managing food-related deficits is very important [6,8,9,10,11]. Most evidence comes from studies of children allergic to cow’s milk. The evidence shows lower dietary calcium intake, slower growth, and reduced bone mineral density in children with cow’s milk allergy [8,9,10]. Similar findings have been reported in children with egg allergy [12]. The influence on growth and nutrient deficiencies is more pronounced in children with multiple food allergies [13,14]. Atopic dermatitis may also affect growth in combination with food allergy or alone [2,14]. Food allergy can also be a common cause of severe, life-threatening reactions in paediatric patients. It can last a lifetime, placing a significant burden on their quality of life and causing other allergic diseases to develop [5].

Nutritional counselling, as a part of food allergy management, has an important role in ameliorating the effects of food allergy on growth and nutrient intake [9,15]. Compared with healthy children, children with cow’s milk, egg, and peanut allergies may have impaired growth. Catch-up growth (catch-up weight and height) is one of the desired outcomes of food allergy management so that their growth can be similar to that of healthy peers from infancy onwards [16].

The influence of egg and peanut allergies on nutritional status is much less studied in the current literature [2]. To our knowledge, peanut allergy alone has not been studied regarding food intake, probably because peanut is not considered an essential food in the European diet.

The present study aimed to determine the long-term effects of allergy to cow’s milk, egg, or peanut on the nutritional status of children compared to their peers without food allergy (control group). We also examined the influence of a single food allergy on their growth and development.

## 2. Materials and Methods

### 2.1. Study Design

A single-centre retrospective cohort study with longitudinal follow-up was conducted at the Department of Allergology, Rheumatology and Clinical Immunology of the University Children’s Hospital of the University Medical Center Ljubljana, Slovenia, from May 2019 to June 2022 (Figure 1). Patients were longitudinally followed at least once yearly from the time the allergy was confirmed until enrolment in the study. The study included three intervention groups (cow’s milk, egg, and peanut allergy) and one control group (healthy children without food allergy but with suspected drug or insect venom allergy). Consecutive patients with a single IgE-mediated food allergy, confirmed by skin prick testing and/or specific IgE in combination with an allergy-focused history, were enrolled. They were divided into three groups: children allergic to cow’s milk only (1), children allergic to eggs only (2), and children allergic to peanuts only (3). Exclusion criteria were non-atopic chronic diseases and a vegetarian or vegan diet. Patients in the control group were enrolled during routine outpatient clinic visits or hospital admissions.

The study was conducted in accordance with the tenets of the Declaration of Helsinki and approved by the Ethics Committee of the Slovenian Ministry of Health. Written informed consent was obtained from all the subjects’ caregivers. According to the standard protocol of our department, upon confirmation of food allergy, all caregivers received written instructions (previously prepared by dietitians and clinicians) on the diet and were regularly followed and managed at the outpatient clinic by the established centre protocol, where diets were checked, and relevant advice given.

Our food allergy management included regular medical follow-up by an allergy specialist (mostly 6-month visits). They were given food allergy education at the time of diagnosis, regular dietary counselling if needed, and additional education about the effects of food allergy. Children with confirmed cow milk allergy were prescribed vitamin D and calcium if not accepting hydrolysed formula or being breastfed. A dietitian was involved in more complex cases and new diagnoses to educate caregivers about dietary management.

Anthropometric measurements were taken regularly during medical follow-up. Their weight and height refer to the data at the time of diagnosis.

We collected data on weight (kg) and height (cm) at the time of confirmation of diagnosis from the patient’s electronic medical record or the patient’s primary paediatrician or caregiver. Data were obtained from the electronic medical records of 10 of 21 (47%) patients with cow’s milk allergy, 20 of 34 (58.8%) children with egg allergy and 21 of 39 (53.8%) children with peanut allergy. For three months, the dietitian collected food diary data (energy intake, protein intake, calcium intake, iron intake, and percentage of daily energy intake from protein, fat, and carbohydrate). On the day of the appointment at the time of enrolment in the study, caregivers brought a 3-day food diary, which the dietitian later analysed. The caregivers later attended the dietitian’s appointment, where they were informed of the results of the food diary analysis and advised on how to improve the diet if necessary. We analysed and compared the diets of the intervention and control groups for the percentage of recommended energy intake, protein intake per kilogram of body weight, percentage of iron and calcium requirements met, and percentage of fat in the diet (percentage of total energy intake).

The mean follow-up time from the allergy diagnosis to the last study visit was 4.85 ± 2.79 years. The children were longitudinally followed from the time of diagnosis; all parents obtained a written dietary and educational program appropriate for their child’s specific food allergy.

### 2.2. Blood Sampling

Blood samples were collected from all enrolled patients at regular outpatient clinic appointments. The blood tests included complete blood count (CBC), iron status (serum iron, transferrin, ferritin, and TIBC level), and serum protein level. All blood samples were analysed in the central biochemistry laboratory at the University Medical Center Ljubljana, Slovenia.

### 2.3. Statistical Analysis

Data were analysed with the R programming language with tidyverse, arsenal, and ggstatsplot packages (R Foundation) [17,18,19,20,21]. Differences between groups (peanut and egg allergy vs. control group) were determined with a *t*-test (numerical data), and a chi-squared test (nominal data) was used to assess differences in frequencies between groups (complete blood count, iron status, serum proteins, anthropological measurements, and food intake analysis). A Mann–Whitney U test (numerical data) and Fisher’s exact test (nominal data) were used to compare cow’s milk allergy and the control group.

Statistical analysis of the retrospective growth data was performed using a *t*-test for the peanut allergy group and a signed-rank test for the cow’s milk and egg allergy group.

The growth parameters of the children with cow’s milk, egg, or peanut allergy were compared with those of the control group. *p* < 0.05 was considered statistically significant. *p*-values were adjusted using the Benjamini-Hochberg and Yekuielti methods. Measures of central tendency were presented as medians and means (SD).

The World Health Organization (WHO) growth charts for nutritional status were used for nutritional assessment and data analysis [22,23].

## 3. Results

In total, 130 children were recruited for the study. Ninety-four children (62 boys and 33 girls) were enrolled with food allergies including 21 patients with cow’s milk allergy, 34 with egg allergy and 39 with peanut allergy, respectively. The mean age at confirmation of food allergy diagnosis was 2.08 ± 1.77 years. The mean age at study enrolment ranged from 2 to 12 years (mean age 6.9 ± 3.2 years). In the control group, 36 children were recruited. This included 19 boys and 17 girls; the mean age ranged from 2 to 12 years (mean age of 8.03 ± 2.79 years). (Table 1).

There were no statistically significant differences in mean z-scores for height for age and weight for age between the groups of children at the study enrolment (between-group comparison for height (*p* = 0.231) and weight for (*p* = 0.381)). The mean z-scores of BMI for age in the cow’s milk allergy group (−0.33 ± 0.68), egg allergy group (−0.12 ± 1.06), the peanut allergy group (0.21 ± 1.37) and the control group (−0.12 ± 1.20) did not show a statistically significant difference between the groups (*p* = 0.410).

Children in the food allergy groups presented clinically with atopic dermatitis (34), asthma (26), hay fever (33), anaphylaxis (2), and angioedema (16). Some children had more than one condition: 3 in the cow’s milk allergy group (14.2%), 9 in the egg allergy group (26.4%), and 8 (20.5%) in the peanut allergy group.

### 3.1. Growth and Nutritional Status in Children with Cow’s Milk Allergy

Children with cow’s milk allergy were meeting their calcium needs with food significantly less than the control group (44.8% vs. 63.8%, *p* = 0.027) (Table 2). Supplementation was not included. Almost all had calcium and vitamin D supplementation at enrolment in the study (90% calcium and 100% vitamin D). The mean age at the diagnosis was 1.58 ± 0.99 years.

Mean carbohydrate intake (as a percentage of energy intake) in the cow’s milk allergy group was also significantly higher than in the control group (57.7% ± 5.8 vs. 53.6% ± 5.8, *p* = 0.048). In the cow’s milk allergy group, this is probably due to a lower fat and/or protein intake (Table 2).

In the cow’s milk allergy group, catch-up growth from diagnosis until enrolment in the study was shown to be statistically significant. Improvements in mean body height for age from −0.252 ± 1.020 Z to 0.161 ± 1.071 Z (*p* = 0.012) were observed (Figure 2). Statistical analysis of body mass index Z scores and weight for age Z scores did not show significant differences from the time of diagnosis of food allergy until enrolment in the study (*p* > 0.05). We could not find any negative impact of food allergy on growth (Figure 2). Statistical analysis of body mass index Z-scores did not show a significant difference from the time of food allergy diagnosis to enrolment in the study (*p* > 0.05).

Comparison of the cow’s milk allergy group to the control group showed a statistically significant difference in the mean age of the children (5.90 ± 3.06 versus 7.9 ± 2.7 years in the control group).

### 3.2. Growth and Nutritional Status in Children with Egg Allergy

No statistically significant differences in energy intake, percentage of calcium intake, percentage of iron intake, and protein intake in grams per kilogram of body weight were found in the egg allergy group compared to the control group (*p* > 0.05).

In the same group, the catch-up growth from diagnosis until enrolment in the study was found to be statistically significant (Figure 3), and improvements were observed in the median Z values for body height for age from −0.288 ± 1.073 to 0.084 ± 0.893 (*p* = 0.017). Statistical analysis of Z values for body mass index and weight for age showed no significant difference from the time of diagnosis of food allergy to enrolment in the study (*p* > 0.05). Egg allergy did not affect the long-term growth of the children.

### 3.3. Growth and Nutritional Status in Children with Peanut Allergy

In the peanut allergy group, no statistically significant differences were observed between the intervention and control groups for energy intake, percentage of calcium intake, percentage of iron intake, and protein intake in grams per kilogram of body weight (*p* > 0.05).

In the same group, the catch-up growth from diagnosis until enrolment in the study was statistically significant, and improvements were observed in the median Z values of body weight for age from 0.183 ± 1.014 Z to 0.415 ± 1.232 Z (*p* = 0.044) (Figure 4). Statistical analysis of body mass index Z value and height for age Z value showed no significant difference from the time of food allergy diagnosis to enrolment in the study (*p* > 0.05). Peanut allergy did not affect the long-term growth of the children.

### 3.4. Laboratory Results

Blood samples were obtained from all 130 participants. Statistical analysis of the laboratory test results did not reveal any statistically significant differences between patients with cow’s milk allergy and the control group (*p* > 0.05).

In patients with egg allergy, we found a statistically significant difference in leukocyte count (8.17 ± 2.1) and mean corpuscular volume (MCV 78.5 ± 3.85, *p* < 0.05) compared to the control group (leukocytes 6.89 ± 1.72, *p =* 0.019; MCV 80.7 ± 3.75, *p* = 0.047). None of these findings were clinically relevant. No significant evidence of increased anaemia compared to the control group was found.

In patients with peanut allergy, we found statistically significant differences in haemoglobin concentration (128.18 ± 7.49) and mean cell haemoglobin concentration (MCHC 349.95 ± 3.85) for the peanut allergy group versus the control group (haemoglobin 123.97 ± 6.722, *p* = 0.025; MCHC 344.76 ± 5.94, *p* = 0.012). None of these findings were clinically relevant.

## 4. Discussion

To our knowledge, this study was the first retrospective, controlled study with longitudinal follow-up to determine the impact of peanut allergy on the nutritional status of children; it was also the first to evaluate the impact of allergy to cow’s milk, egg, and peanut on nutritional status in children from Southern and Eastern Europe. Growth in peanut allergy has not yet been studied, mainly because peanuts are not considered an essential food. Our study provides important insights into the nutritional status of children with food allergies and the impact of early medical and dietary interventions.

Food allergy represents a significant burden for affected children and their caregivers, which can continue into adulthood [24]. The burden can also be psychological and socioeconomical. In our centre, we often meet children with food allergy with anxiety about eating certain foods, uncertainty about eating outside of their home and also eating disorders. Our cases also include children and their worried carers with a huge burden of food allergies and the cost of replacement foods.

In all of the food allergy groups we studied, weight for age improved from the time of diagnosis to the time of enrolment in the study. This is the opposite of what has been found in the literature, where dietary concerns were quite common. Most evidence shows that, in food allergy groups, children are more often undernourished and stunted [25,26].

For atopic dermatitis, previous studies have shown a negative influence on growth and nutrient intake. Their growth was also shown to be worse [1,14]. It is a self-limited disease that very rarely affects children with good management of food allergy or lasts for a very long time. The present study found no negative effects of atopic dermatitis on growth; children with or without atopic dermatitis had the same growth potential. We believe that the likely cause of these positive results is the high level of nutritional awareness that has increased over the last decade. Thorough education of our caregivers and strict management of food allergies are also very important. Regular education of patients and their carers by our hospital teams (doctors, nurses, and dieticians) starts at the time of diagnosis of food allergy and continues regularly throughout its management. One of the important factors is also the availability of many free food allergy educational resources on the internet. Some support is provided by our parent interest groups. Increased awareness of the impact of allergies on growth is crucial. There is now a wide range of foods available for people with food allergies and their availability is increasing. Additionally, the role of a dietitian in this area is essential [9,15], which is why our practice offers regular dietary consultations. Dietary counselling can also help educate caregivers about how best to replace allergens and minimise the impact of food allergy on dietary intake, growth and development of children with food allergy. Previous studies suggest there is still a concern regarding calcium intake for individuals with cow’s milk allergy [1,8]. In the study, there was a statistically significant lower calcium intake compared to the control group. In our clinical practice, we aim to address calcium deficiency through various means, such as supporting breastfeeding, increasing the intake of appropriate formulas or vegetable drinks fortified with calcium, and calcium supplementation as another resort. The majority of patients rely on calcium supplementation. Insufficient data prevented us from demonstrating the effect of reduced calcium intake on bone density. We collected data on bone mineral density from only 9 out of 21 patients with cow’s milk allergy, and all showed normal bone mass density. Conducting statistical analysis was not feasible. Previous studies in indicated lower bone mass density in prepubertal children with cow’s milk allergy [27].

The average consumption of carbohydrates (as a percentage of energy intake) was significantly greater in the cow’s milk allergy group compared to the control group. This is likely due to a reduced fat and/or protein intake in the cow’s milk allergy group. It is the first discovery of its kind in the literature, implying that fat and fat-soluble vitamin consumption may be lower in this population. On the other hand, their intake of simple sugars could be higher and this could have a negative impact on their health.

Contrary to previous research studies [1,4,13,26], our study demonstrated that children with cow’s milk allergy were able to achieve normal growth and development without any notable impact. Normal growth may be due to increased awareness and education about cow’s milk allergy and its effects. In our practice, we encourage breastfeeding and provide regular education to our patients once an allergy has been diagnosed, taking appropriate measures as necessary to minimize the potential negative effects. In addition, a wide range of hypoallergenic formulas and vegetable drinks with added calcium are readily available. Our practice always discourages the use of vegetable drinks without added calcium and the usage of hydrolyzed vegetable formulas.

There was no statistically significant difference in dietary energy and nutrient intake between the egg allergy group and the control group (*p* > 0.05). Contrary to previous research, our study suggests that children with egg allergy can achieve normal growth and development if good education is provided to caregivers [1,4,26,28]. Proper education about egg allergy is very important to know how to replace eggs with other protein-rich foods. The food and nutrient intake among children with egg allergy was comparable to that of the control group and no significant difference was observed (*p* > 0.05). In contrast to previous studies, our findings indicate no increased occurrence of anaemia in children with egg allergy compared to the control group [11]. In the case of egg allergy, we always recommend increasing the intake of meat, which is probably the reason for the good results.

The peanut allergy group had no statistical difference in dietary energy and nutrient intake compared to the control group (*p* > 0.05). To our knowledge, there has been no study on how a single peanut allergy affects growth and nutrient intake. Previous studies have mainly focused on the early introduction of peanuts and preventing future allergies. This is shown to be effective in preventing peanut allergy later in life [29,30]. Our study found no significant impact of the peanut allergy on growth. No statistically significant differences in the blood laboratory parameters were found in children with peanut allergy compared to the control group. Research on this topic is lacking, likely because peanuts are not considered an essential food. On the other hand, early introduction of peanuts will hopefully prevent allergy and the prevalence of peanut allergy should decrease in the future.

The control group in the study also had significant nutritional deficiencies, and the dietary advice probably had a positive effect on their nutritional intake as well. It was very important to compare the cow’s milk, egg and peanut allergy group with them, because we found, in many cases, that their nutritional needs were not optimally met, even in the group of children without food allergy. Their nutritional needs should also be met optimally for better growth.

A potential limitation of our study is that certain weight and height data at the time of diagnosis were reported by children’s caregivers, which could potentially result in inaccuracies. Another limitation is the retrospective design employed, which may have introduced confounding variables. It was also not possible to rule out potential errors in dietary intake reporting by caregivers of children with food allergies or control participants. However, it is unlikely that this error would differ between children in all the groups. In addition, our study has the limitation of lacking follow-up data for children in the control group. The question of the starting point is when to start follow-up, as there was a significant range of age at diagnosis confirmation in the food allergy groups.

Another limitation of our study is that it was a short-term study in a food-allergic population. A long-term study might shed more light on the effect of diet on growth in people with food allergies. Even when their allergies have resolved, many patients remain on a restricted diet and may still have growth or nutritional problems. Furthermore, we encountered the significant drawback of insufficient data regarding mineral bone density in children with cow’s milk allergy, and statistical analysis was not feasible. The study and control groups differed in some respects that could influence the results. Children in the cow’s milk allergy group were considerably younger than children in the control group. The likely reason is that cow’s milk allergy is usually diagnosed and resolved earlier in life.

## 5. Conclusions

In conclusion, our study showed that allergies to cow’s milk, eggs, or peanuts do not directly impact an individual’s nutritional status. Children with food allergies were able to achieve growth similar to their peers with no food allergies. To achieve the best results, a clinical care for children with food allergies should include patient-tailored dietary counselling with written instructions for child’s caregivers and regular outpatient visits with an allergy specialist. The child’s caregivers should receive also comprehensive information about the possible negative effects of food allergies on growth and nutrient intake as well as information on appropriate food substitutes to avoid the deficits. In managing food allergies, these actions can provide optimal care and ensure normal growth and development of children with cow’s milk, egg, or peanut allergies.

## Figures and Tables

**Figure 1 nutrients-16-00048-f001:**
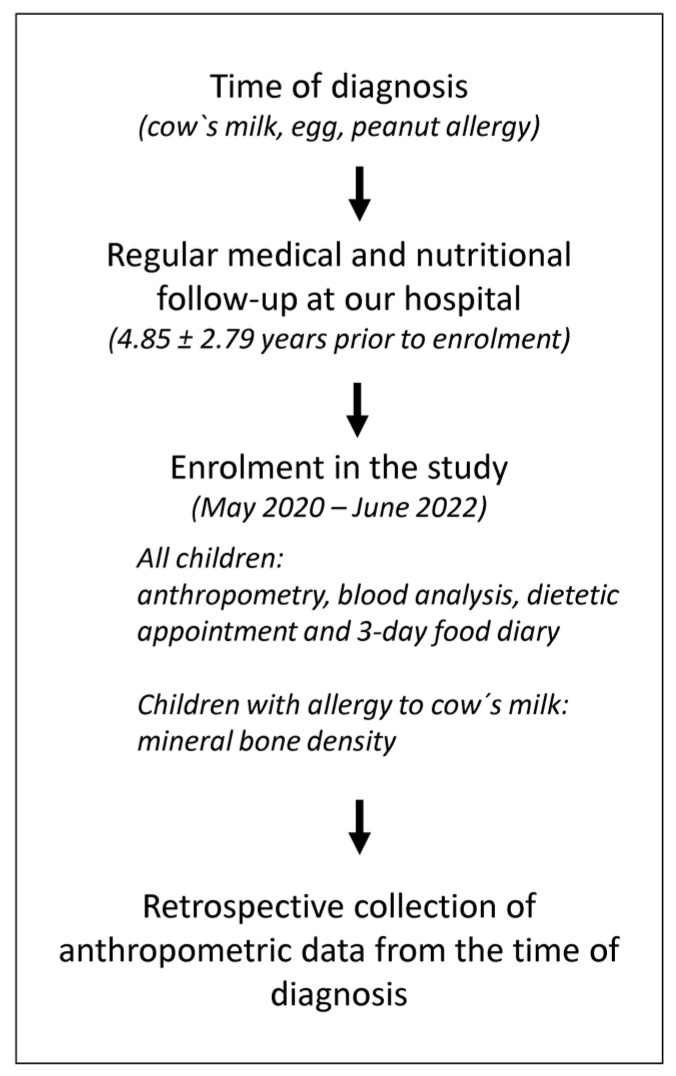
Study design.

**Figure 2 nutrients-16-00048-f002:**
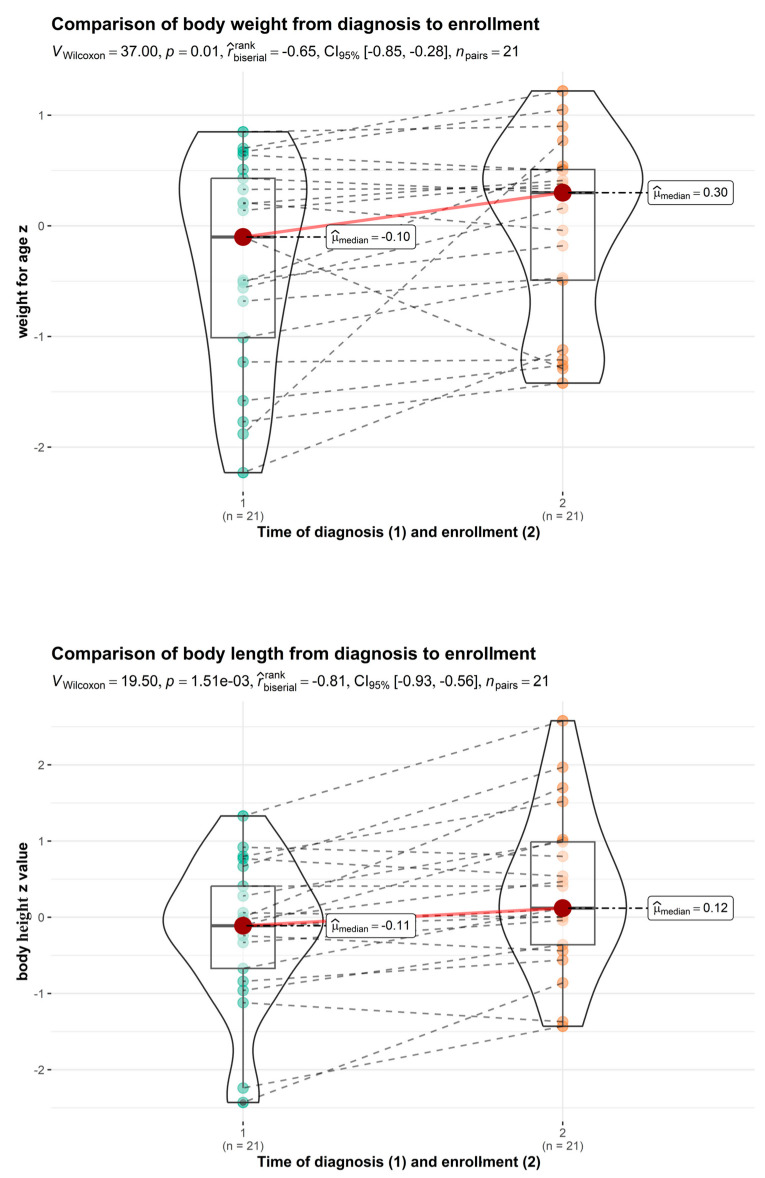
Comparison of weight for age and height for age (cow’s milk allergy group, time of diagnosis vs. enrolment).

**Figure 3 nutrients-16-00048-f003:**
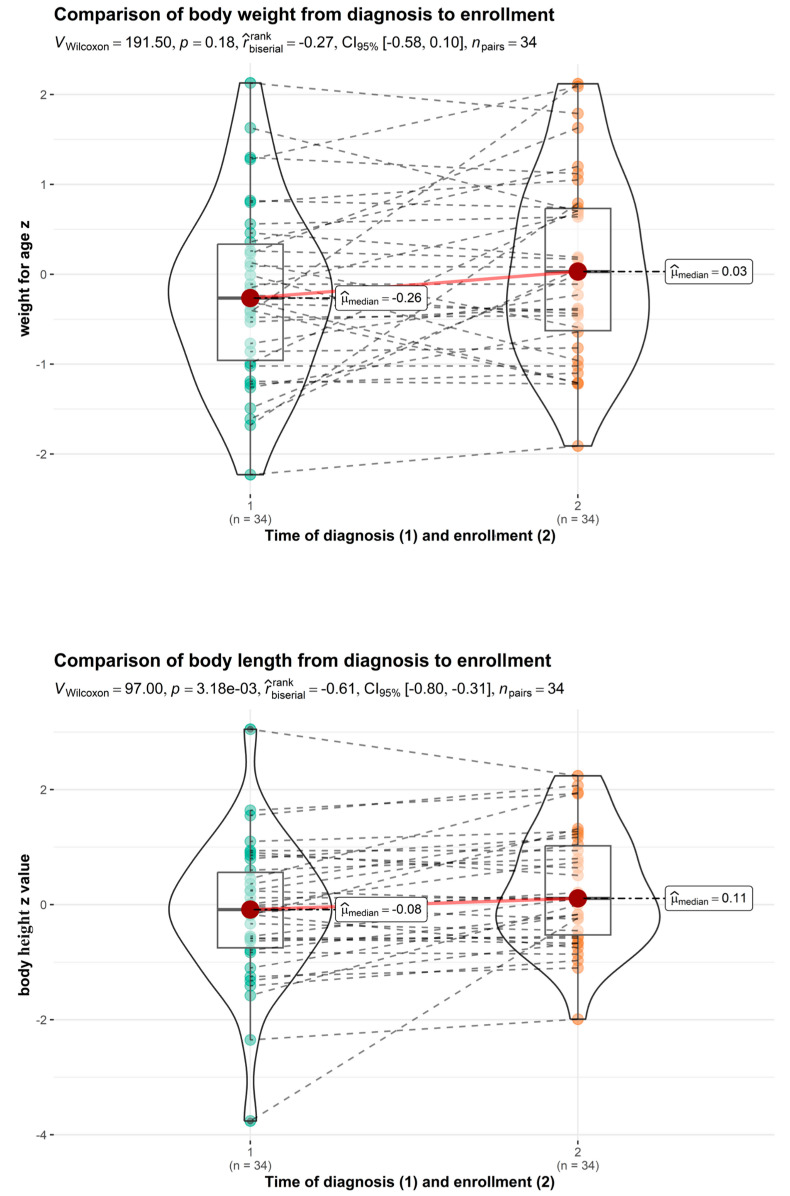
Comparison of weight for age and height for age (egg allergy group, time of diagnosis vs. enrolment).

**Figure 4 nutrients-16-00048-f004:**
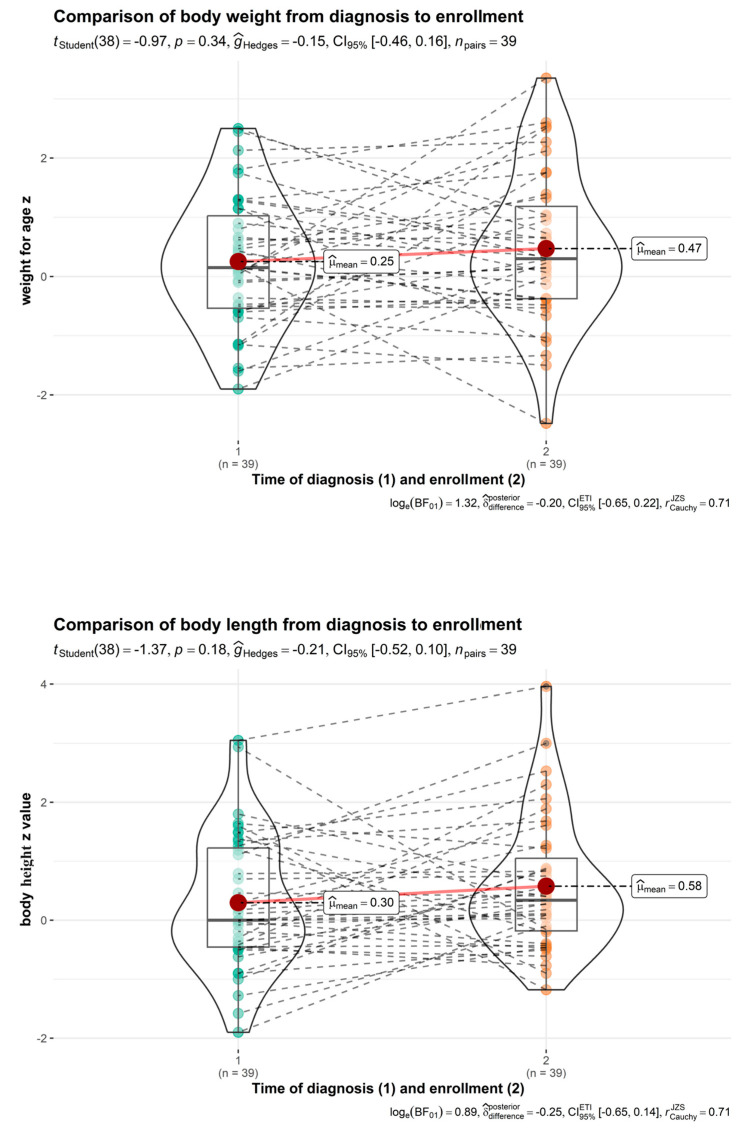
Comparison of weight for age and height for age (peanut allergy group, time of diagnosis vs. enrolment).

**Table 1 nutrients-16-00048-t001:** Patients’ characteristics.

All Participants (*n* = 130)	Cow’s Milk (*n* = 21)	Egg (*n* = 34)	Peanuts (*n* = 39)	Control Group (*n* = 36)
Age (years) at diagnosis (SD)	1.58 (0.99)	1.41 (1.12)	2.88(2.11)	/
Age (years) at enrolment (SD)	5.90 (3.06)	6.24 (3.06)	8.08 (2.99)	8.03 (2.79)
Sex (male)	14 (66.7%)	20 (55%)	25 (64%)	18 (48.7%)
Body weight for age z at diagnosis	−0.254 (0.866)	−0.347 (0.838)	0.183 (1.014)	/
Body height for age z at diagnosis	0.252 (1.020)	−0.288 (1.073)	0.229 (1.092)	/
Body weight for age z at enrolment	−0.02 (0.83)	0.09 (1.03)	0.47 (1.26)	0.35 (0.97)
Body height for age z at enrolment	0.34 (1.05)	0.27 (1.01)	0.58 (1.13)	0.74 (1.04)
Body mass index z at enrolment	−0.33 (0.68)	−0.12 (1.06)	0.21 (1.37)	−0.12 (1.20)
Comorbidities				
Atopic dermatitis (*n*, %)	7 (33%)	14 (41.1%)	15 (38.4%)	5 (13.8%)
Asthma (*n*, %)	6 (28.5%)	8 (23.5%)	12 (30.7%)	4 (11.1%)
Hay fever (*n*, %)	4 (19%)	11 (32.3%)	18 (46.1%)	5 (13.8%)
Anaphylaxis (*n*, %)	0	0	2 (0.05%)	3 (8.3%)
Angioedema (*n*, %)	0	4 (11.7%)	12 (30.7%)	2 (5.5%)
Dietary intake				
Coverage of energy needs (%)	88.73 (22.97)	86.40 (25.87)	86.70 (22.86)	89.59 (26.97)
Coverage of iron needs (%)	77.39 (28.22)	68.16 (25.87)	71.76 (25.80)	77.92 (38.82)
Coverage of calcium needs (%)	44.86 (22.89)	70.01 (32.10)	65.21 (27.02)	63.83 (25.69)
Protein intake (g/kg)	2.31 (0.88)	2.30 (0.91)	2.17 (0.82)	2.20 (0.89)
Fat intake (% daily energy)	27.48 (4.61)	31.09 (5.34)	31.85 (5.37)	30.30 (5.03)
Carbohydrate (% daily energy)	53.65 (5.80)	53.68 (6.66)	51.97 (6.69)	53.65 (5.80)
Vitamin D supplementation (*n*, %)	21 (100%)	/	/	/
Calcium supplementation (*n*, %)	19 (90%)	/	/	/

Legend: standard deviation (SD).

**Table 2 nutrients-16-00048-t002:** Calcium needs coverage (%) and carbohydrate intake in the cow’s milk allergy and control group.

**Calcium Need Coverage (%)**	**Control**	**Cow’s Milk**	**Total**	** *p* **
Mean (SD)	63.83 (25.69)	44.86 (22.89) *	7.26 (3.05)	0.027 *
Median (Q1, Q3)	61.91 (46.56, 74.55)	45.60 (29.20, 53.50) *	53.89 (38.10, 70.40)	
Min–Max	13.89–124.67	17.55–118.83	13.89–124.67	
**Carbohydrate (% Energy Intake)**	**Control**	**Cow’s Milk**	**Total**	** *p* **
Mean (SD)	53.65 (5.80)	57.76 (5.80) **	55.14 (6.09)	0.048 **
Median (Q1, Q3)	61.91 (46.56, 74.55)	57.00 (53.00, 63.00)	54.00 (51.00, 58.00)	
Min–Max	44.00–76.00	49.00–66.00	44.00–76.00	

Legend: standard deviation (SD), interquartile range (Q1, Q3), *, **—statistically significant.

## Data Availability

The data that support the findings of this study are available from the corresponding author upon reasonable request.

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
