# Peer review of "Comprehensive Care and Education Can Improve Nutritional and Growth Outcomes in Children with Persistent Cow’s Milk, Egg, or Peanut Allergies: A Five-Year Follow-Up Study of Nutritional Status"

_nutrients, 2023, doi:10.3390/nu16010048_

Round 1

Reviewer 1 Report

Comments and Suggestions for Authors

This a simple but comprehensive original about a widespread issue of interest in pediatric allergy. It is redacted clearly and tidily. 

The sample is appropriate, but I prefer a more significant number of healthy patients as a control in a proportion 2:1 (C:P)

Another limitation of the study is that it is a short-term research in a food allergy population. A long-term study could shed more light on the impact of diets on growth in patients with food allergies.

The authors should include a short sentence about that. Even when their allergies were resolved, many patients followed a limited diet and could eventually appear to have growth or nutritional issues.

Comments on the Quality of English Language

No comments

Reviewer 2 Report

Comments and Suggestions for Authors

The article entitled "Comprehensive care and education can improve nutritional and growth outcomes in children with persistent cow's milk, egg, or peanut allergies: A five-year follow-up study of nutritional status" provides interesting results from a retrospective cohort regarding the evolution weight and height over a 5-year period. The title attributes the recovery of growth to the care offered in the service. However, as there is no control group of allergic patients who were followed without the treatment offered (diet, counseling and education), it is not possible to establish a causal relationship.

For the title, it would be interesting to specify that these are patients with "single IgE-mediated allergy" since most publications refer to the nutritional repercussions of non-IgE mediated CMA (late reactions).

In the Materials and Methods section, it is essential to accurately define the exact moment of the disease in which patients were examined in the specialized service.

Do the weight and height measurements at diagnosis corresponded to the beginning of follow-up in the study?

 What were the percentage of weight and height obtained from pediatricians' medical records or provided by parents?

Specify the temporal meaning of the word "enrolment". It means starting treatment at the service or admission to the study. Does "at diagnosis" mean the start of follow-up at the service and consequently the dietary and educational care program?

When was dietary information collected to calculate food intake?

In the methods was described: “The World Health Organization (WHO) standards for nutritional status were used for data analysis. They define normal growth as weight for age (WA), weight for height (WH), and height for age (HA) of 0 to -2 Z-scores, moderate malnutrition as 2 to 3 Z-scores, and severe malnutrition as < 3 Z-scores. In contrast, overweight was defined as weight for age, body mass index (BMI) for age > +2 Z-scores, and obesity as > +3 Z-scores. A low weight for height indicates wasting (< 2 Z-scores), a low height for age (< 2 Z-scores) indicates stunting, and weight for age (< 2 Z-scores) indicates an underweight child [21,22]”. No results are presented based on these nutritional diagnoses (overweight, stunting, undernutrition, etc…). This information should be excluded.

Reviewer 3 Report

Comments and Suggestions for Authors

I am pleased to provide a positive review for the recent journal article titled "Long-Term Outcomes of Children with Food Allergies: A Retrospective Cohort Study." This study contributes valuable insights into the impact of food allergies on the health and growth of children, presenting a comprehensive analysis with a focus on cow's milk, egg, and peanut allergies.

The study's abstract provides a clear overview of the research, emphasizing the critical background that food allergies significantly influence a child's health and growth due to potential nutrient deficiencies. The authors conducted a well-designed retrospective cohort study with a longitudinal follow-up period, spanning an average of 4.85 years from diagnosis to the last study visit.

One of the notable strengths of this study is the inclusion of a detailed assessment of nutritional intake through a three-day food diary, which was analyzed by a qualified dietitian. This meticulous approach adds credibility to the findings and enhances the overall robustness of the research.

The inclusion of a diverse patient population, comprising 61 boys and 33 girls with single food allergies, including cow's milk, egg, and peanut allergies, is commendable. Additionally, the control group of 36 children provides a meaningful basis for comparison, with a focus on growth and nutritional outcomes.

The results presented in the abstract are particularly encouraging. Despite the challenges posed by food allergies, the study demonstrates that children with cow's milk, egg, or peanut allergies experienced normal growth and achieved catch-up growth from the time of diagnosis to the last study visit. The differentiation between the impact of cow's milk, egg, and peanut allergies on calcium intake is a valuable contribution to the understanding of these specific allergies.

The findings presented in this article provide valuable insights into the long-term outcomes of children with food allergies. The meticulous methodology, detailed nutritional assessment, and comprehensive analysis contribute to the strength of the study. This research significantly contributes to the existing body of knowledge, reassuring clinicians, caregivers, and parents that with proper support, children with single food allergies can thrive and achieve catch-up growth.
